# Transformers are efficient hierarchical chemical graph learners

**Zihan Pengmei**[†]
Department of Chemistry
The University of Chicago
Chicago, IL 60637

**Zimu Li**[†]
Yau Mathematical Sciences Center
Tsinghua University
Beijing 100084, China

**Chih-chan Tien**
Department of Computer Science
The University of Chicago
Chicago, IL 60637

**Risi Kondor**
Department of Computer Science
The University of Chicago
Chicago, IL 60637

**Aaron R. Dinner**[*]
Department of Chemistry
The University of Chicago
Chicago, IL 60637

## Abstract

Transformers, adapted from natural language processing, are emerging as a leading approach for graph representation learning. Contemporary graph transformers often treat nodes or edges as separate tokens. This approach leads to computational challenges for even moderately-sized graphs due to the quadratic scaling of self-attention complexity with token count. In this paper, we introduce SubFormer, a graph transformer that operates on subgraphs that aggregate information by a message-passing mechanism. This approach reduces the number of tokens and enhances learning long-range interactions. We demonstrate SubFormer on benchmarks for predicting molecular properties from chemical structures and show that it is competitive with state-of-the-art graph transformers at a fraction of the computational cost, with training times on the order of minutes on a consumer-grade graphics card. We interpret the attention weights in terms of chemical structures. We show that SubFormer exhibits limited over-smoothing and avoids over-squashing, which is prevalent in traditional graph neural networks.

## 1 Introduction

Many systems ranging from social networks to molecular structures involve interactions between discrete elements and thus can be described by graphs. It remains challenging to identify the features of these systems that best enable learning their properties from data. Manually choosing features is subjective, and the computational cost of kernel-based methods scales cubically with the size of the dataset; kernel methods are also hard to parallelize, in contrast to neural network methods [51]. Graph neural networks (GNNs), in which graph structures directly define model structures that are inherently sparse and, in turn, efficient, have emerged as attractive alternatives [16]. However, GNNs are not the only neural-network architectures that show promise for graph-structured data learning. Transformers [47] also appear to be able to learn graphs just as they are able to learn the semantic

---

[*]Correspondence author. Email: dinner@uchicago.edu

Preprint. Under review.

structures of sentences by modulating the weights of a fully-connected architecture [11, 7, 56, 31, 43]. The stark contrast between these two architectures with respect to structural inductive bias calls for better understanding of how they encode graph-structured data and how they can be combined to advantage.

In the present paper, we consider the case of learning molecular properties from chemical structures. A challenge that arises in this case is that nodes (atoms or functional groups) separated by many edges (bonds) can interact owing to the delocalized nature of electronic structure and the arrangement of the atoms in space. Prevailing graph learning methods are based on message-passing (MP) neural networks (NNs) [16]. In each layer of an MPNN, nodes aggregate information from their nearest neighbors. Capturing long-range interactions requires repeated exchanges between neighbors and thus many MP layers. MPNNs with insufficient numbers of layers exhibit *under-reaching*, in which nodes remain oblivious to information associated with nodes beyond a certain number of hops [2, 46]. While this problem can be solved by stacking layers, as in relatively simple MPNNs like GCN [28] and GAT [48], this introduces two other issues: *over-squashing* and *over-smoothing* [2, 46, 8, 29]. The former refers to the insensitivity of a feature vector at a node to variations in feature vectors at distant nodes due to excessive compression of information by repeated MP. Over-smoothing refers to feature vectors across nodes becoming increasingly uniform after numerous MP iterations.

Graph transformers, with their ability to allow tokens of atom features to interact directly through a fully-connected structure, are posited to address the over-squashing issue. However, they are not without their own set of challenges. These include a computational cost that scales quadratically with the number of nodes [47] and persistent concerns about over-smoothing [9, 44]. Furthermore, the performance of graph representation learning models is generally assessed in terms of their ability to separate non-isomorphic graphs [54, 3], and, like most efficient models [37, 38], pure graph transformers cannot separate graphs that are indistinguishable under the 1-Weisfeiler-Lehman test (1-WL) [27], an algorithm based on color refinement of graph nodes [50].

In this work, we introduce the Subgraph Transformer (SubFormer), a novel molecular graph learning architecture that combines the strengths of MPNNs and transformers to address the challenges highlighted earlier. Central to our approach is the decomposition of molecular graphs into coarse-grained representations using hierarchical clustering methods. We adopt the concept of junction trees, as proposed in [23], which can be likened to a "molecular backbone", where each node represents a subgraph or cluster derived from the original graph. SubFormer operates in two distinct phases. Initially, the graph-level features of the molecule are locally aggregated using a shallow MPNN. This approach avoids both the over-squashing and over-smoothing problems associated with deeper MPNNs. Following this, the resulting coarse-grained features, representing substructures within the molecule, are passed through a standard transformer. The transformer's direct interactions between tokens eliminate the need for numerous iterations to access long-range interactions, effectively addressing both the under-reaching problem and the over-squashing problem. By clustering nodes within the graph, we achieve a computational cost reduction for the transformer, proportional to a power of the average node count per cluster. Notably, our framework offers an expressive power that surpasses the 1-WL test.

This paper is organized as follows. Section 2 provides background on MPNNs and transformers, and we discuss pertinent models, such as the graph transformer[47, 11, 40, 27] and the junction tree variational autoencoder [23]. In Section 3, we present the update rule of the Subgraph Transformer (SubFormer) and then experimental results; we show that SubFormer performs on par with leading graph transformers on standard graph benchmarks, including the ZINC [22], long-range graph benchmarks [12], and MoleculeNet datasets [52]. We summarize in Section 4. Theoretical analysis for the expressive power of SubFormer, further details and analysis of the benchmarks, and timing information are provided in the Appendices.

## 2   Background

**Message passing graph neural networks.** A graph $G$ consists of a collection $V(G)$ of $n$ nodes and a collection $E(G)$ of edges connecting selected pairs of nodes. It is conventional to express the graph by its adjacency matrix $A$ with elements $a_{ij}$, where $a_{ij} = 1$ if nodes $i$ and $j$ are connected, and $a_{ij} = 0$, otherwise. Let $D$ denote the diagonal matrix where $d_{ii}$ is the number of nodes connected to $i$. Then the normalized graph Laplacian is defined as $L = I - \sqrt{D^{-1}} A \sqrt{D^{-1}}$ [11, 27]. We

take its eigenvectors as the positional encoding of the transformer that we introduce later. Let $\{x_i\}_{i \in V(G)}, \{y_{ij}\}_{(i,j) \in E(G)}$ be initial node and edge features. Then a MPNN [16] updates features according to the following recursion formulas:

$$x_i^{(0)} = x_i; \quad x_i^{(l+1)} = \Psi\Big(x_i^{(l)}, \text{AGG}_{j \in N(i)} \Phi(x_i^{(l)}, x_j^{(l)}, y_{ij})\Big), \tag{1}$$

where AGG is a function that aggregates node and edge features in the neighborhood $N(i)$ of node $i$ by summation and $\Psi$ and $\Phi$ are trainable functions implemented as neural-network layers.

**Graph transformers.** A transformer layer [47] is mathematically represented by a parameterized function $f^{(l)} : \mathbb{R}^{m \times d} \to \mathbb{R}^{m \times d}$ with $m$ tokens $z_i \in \mathbb{R}^d$, where $d$ is the dimension of the feature vector $x_i$. The tokens $z_i$ are updated through multi-head self-attention (MHSA) at the $l$-th layer:

$$a_{h,ij}^{(l)} = \text{softmax}\Big(\frac{\langle Q_h^{(l)}(z_i^{(l)}), K_h^{(l)}(z_j^{(l)}) \rangle}{\sqrt{k}}\Big); \quad u_i^{(l)} = \sum_{h=1}^{H} W_h^{(l)} \sum_{j=1}^{n} a_{h,ij}^{(l)} V_h^{(l)}(z_j^{(l)}), \tag{2}$$

where $Q_h^{(l)}$, $K_h^{(l)}$, and $V_h^{(l)} \in \mathbb{R}^{k \times d}$ are the query, key and value matrices, and $W_h^{(l)} \in \mathbb{R}^{d \times k}$ is a weight matrix per head. The head number is $H = n/k$, and the softmax function is used to produce a sequence of discrete probability distributions known as attention weights, $a_{h,ij}$. The output of the MHSA is normalized [34, 53] and passed through a fully connected layer, the output of which is again normalized:

$$w_i^{(l)} = \text{LayerNorm}(z_i^{(l)} + u_i^{(l)}); \quad z_i^{(l+1)} = \text{LayerNorm}(w_i^{(l)} + W_2\sigma(W_1 w_i^{(l)})), \tag{3}$$

where $\sigma$ is a non-linear activation function such as ReLU.

Graph transformers [11, 31, 56, 43, 27, 58] adapt the transformer architecture to studies of graphs by incorporating positional or structural encoding as a soft inductive bias. For instance, one could either add or concatenate the Laplacian eigenvectors to the initial tokens [11, 31, 49, 27], or one could use the shortest path between nodes $i$ and $j$ to bias the self-attention weights $a_{h,ij}$ [56, 59]. In this paper, we add the eigenvectors of the graph Laplacian and the shortest path matrix to tokens without making any further modifications to the MHSA in a standard transformer architecture.

**Junction tree variational autoencoders.** The framework that we describe below employs both GNN and transformer architectures to study local structure of the original graph as well as long-range interactions through a hierarchically coarse-grained graph. We decompose molecular graphs based on chemically meaningful local structures like rings and bonds following the strategy in [23]. To be specific, given a molecular graph $G$, its junction tree is crafted as follows. (1) We search all rings and bonds (edges) which are not contained in a ring from the graph and record them as clusters $C_i$. (2) If an atom belongs to more than two clusters, we remove it from the existing clusters and make a new cluster containing only that atom. (3) We draw a new graph $G'$ with the $C_i$ as nodes. Two clusters $C_i$ and $C_j$ are joined by an edge if their intersection is nonzero in $G$. (4) We take a maximal spanning tree $T$ of $G'$. Some examples are presented in Figs. 1, 2, 8, and 7.

## 3   Results

**SubFormer approach.** To deal with the aforementioned issues, we propose Subgraph Transformer (SubFormer). SubFormer consists of two parts. The first is characterized by the subgraph-to-token routine, which employs a few MP layers to gather local information and compress it into a coarse-grained graph using a graph decomposition scheme. We follow [15] and use the junction tree autoencoder [23] mentioned above. To be specific, suppose $U = (u_{\mu i}), \mu, i = 1, ..., m$ is a matrix whose rows are eigenvectors of either Laplacian or shortest path distance matrix, where the Greek indices $\mu$ represent eigenvalues. We first take column vectors $a_i^{\text{PE}} = (u_{i\mu})$, and then let $X \in \mathbb{R}^{n \times d}$ denote the feature matrix of the molecular graph $G$ and let $Z \in \mathbb{R}^{m \times d}$ denote the feature matrix of the junction tree $T$. We initialize node features by concatenating $x_i$ from $X$ for each node $i$ with the positional encoding $a_i^{\text{PE}}$ and then applying a weight matrix $W_1$:

$$x_i^{(0)} = W_1[x_i, a_i^{\text{PE}}] \quad \text{for} \quad i = 1, ..., n. \tag{4}$$

As a caveat, $a_i^{\text{PE}}$ consists of the $i$th components from all eigenvectors we taken from SVD to ensure the permutation equivariance. Moreover, since SVD cannot be unique due to eigenvector sign flips or

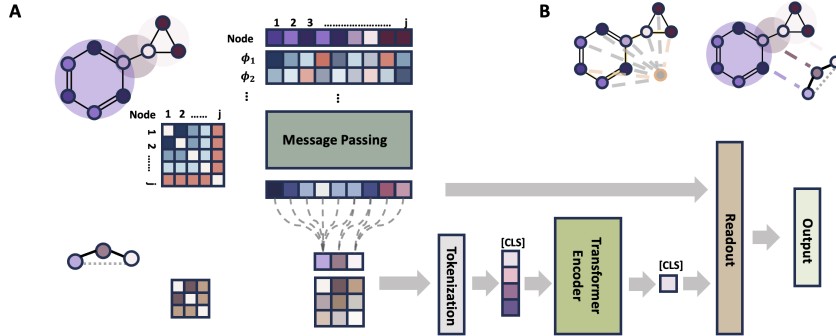

Figure 1: Architectures. (A) Example molecule, its graph Laplacian, and the SubFormer architecture with dual readout from hierarchical graphs. $\phi_i$ denotes the eigenvectors of the graph Laplacian or shortest path matrix. [CLS] refers to an classification token used in transformer encoders for information aggregation. (B) Illustration of the virtual node trick (left) and graph coarse-graining (right). The virtual node decreases the shortest path between any two nodes to two hops at most.

repeated eigenvalues, we follow [30] by adding weight matrix $W_1$ which could possibly learn proper linear combinations of eigenvectors during the training. We also apply SignNet proposed in [35], which input both $\pm a_i^{\mathrm{PE}}$ through one non-linear activation to cancel out the sign flip freedom.

We then compress the local information of $G$ into $T$ using the aforementioned local message-passing mechanism with a fully-connected layer as follows:

$$
\begin{aligned}
X^{(l)'} &= \mathrm{MPNN}(X^{(l)}); \\
X^{(l+1)} &= X^{(l)'} + \theta_1 \sigma(S^T Z^{(l)} W_2^{(l)}); \\
Z^{(l+1)} &= Z^{(l)} + \theta_2 \sigma(S X^{(l+1)} W_3^{(l)}),
\end{aligned}
\tag{5}
$$

where $\theta_i$ and $W_i$ are trainable weights, and we denote by $S \in \{0,1\}^{|V(T)| \times |V(G)|} = \{0,1\}^{m \times n}$ the matrix that assigns nodes from $G$ to clusters and thus nodes of $T$. Updated node features $X_{\mathrm{out}} = X^{(L)}$ are added together to obtain $x_{\mathrm{out}}$. Then we tokenize $Z_{\mathrm{out}} = Z^{(L)}$ composed of vectors $z_i$ representing node features by applying a weight matrix $W_4$ to its concatenation with the positional encoding $b_i^{\mathrm{PE}}$ of the coarse-grained tree analogous to Eq.(4) to preserve permutation equivariance on the tree:

$$
z_i^{(0)} = W_4[z_i, b_i^{\mathrm{PE}}] \quad \text{for} \quad i = 1, ..., m.
\tag{6}
$$

It is also a common option to attach a classification token $z_0^{(0)}$ of learnable parameters [10, 57] ([CLS] in Fig. 1). After training a standard Transformer of $L$ layers to learn subgraph-level information, we read out the class token concatenated with $x_{\mathrm{out}}$ optionally as

$$
z_{\mathrm{out}} = [z_0^{(L)}, x_{\mathrm{out}}].
\tag{7}
$$

The advantages of our SubFormer architecture are discussed in conjunction with our experimental results below. We show that SubFormer is more powerful than the 1-WL algorithm for the graph isomorphism problem [50, 39, 26] in Appendix A.

**Predicting molecular properties.** We first learn and predict octanol-water partition coefficients using the ZINC dataset [22]. The graphs are defined by the non-hydrogen atoms (nodes) and their bonds (edges); the node features are the atom types, and the edge features are the bond types. Our results are competitive with those of a state-of-the-art graph transformer, GraphGPS, and surpass those of other graph transformers (Table 1). We also obtain results comparable to GraphGPS (as well as GT and SAN) for the Peptides-struct benchmark, a long-range graph benchmark [12] (Table 2).

The MoleculeNet datasets target predictions of molecular properties such as toxicity, solubility, and bioactivity [52]. We split the dataset and evaluate performance following [52]. SubFormer exhibits strong performance across these diverse tasks. Notably, a dual readout approach [15], which integrates information from both the original graph and the coarse-grained tree, improved prediction accuracy in most (but not all) cases (Tables 1 and 3).

Table 1: Results for the ZINC dataset. Lower values are better. Scores for published architectures are taken from [43, 45, 15, 56, 30].

| Model | MAE |
|-------|-----|
| GCN | 0.367±0.011 |
| GAT | 0.384±0.007 |
| GIN | 0.408±0.008 |
| HIMP | 0.151±0.006 |
| AUTOBAHN | 0.106±0.004 |
| GT | 0.226±0.014 |
| SAN | 0.139±0.006 |
| Graphormer | 0.122±0.006 |
| GraphGPS | **0.070±0.004** |
| Ours(580k) | 0.094±0.003 |
| Ours(dual readout,slim,200k) | 0.084±0.004 |
| Ours(dual readout,VN, 567k) | 0.078±0.003 |
| Ours(dual readout,567k) | 0.077±0.003 |

Table 2: Results for the Peptides-struct dataset of long-range graph benchmarks. Lower values are better. Scores for published architectures are taken from[43].

| Model | MAE |
|-------|-----|
| GCN | 0.3496±0.0013 |
| GIN | 0.3547±0.0045 |
| GT | 0.2529±0.0016 |
| SAN | 0.2545±0.0012 |
| GraphGPS | 0.2500±0.0005 |
| Ours(567k) | **0.2464±0.0012** |

Table 3: Results for the MoleculeNet datasets. Higher values are better. *Because published GraphGPS results were not available for these datasets, we conducted our own experiments; because GraphGPS overfits the MUV dataset, we report the best value in parentheses as well. Scores for published architectures are taken from [42, 13, 52, 15, 45].

| Dataset | TOX21 | TOXCAST | MUV | MOLHIV |
|---------|-------|---------|-----|--------|
| Num. Task | 12 | 617 | 17 | 1 |
| Metic | ROC-AUC | ROC-AUC | AP | ROC-AUC |
| Random Forest | 0.769±0.015 | N/A | N/A | 0.781±0.006 |
| XGBoost | 0.794±0.014 | 0.640±0.005 | 0.086±0.033 | 0.756±0.000 |
| Kernel SVM | 0.822±0.006 | 0.669±0.014 | 0.137±0.033 | 0.792±0.000 |
| Logistic Regression | 0.794±0.015 | 0.605 ± 0.003 | 0.070±0.009 | 0.702±0.018 |
| GCN | 0.840±0.004 | 0.735±0.002 | 0.114±0.029 | 0.761±0.010 |
| GIN | 0.850±0.009 | 0.741±0.004 | 0.091±0.033 | 0.756±0.014 |
| HIMP | **0.874±0.005** | 0.721±0.004 | 0.114±0.041 | 0.788±0.080 |
| AUTOBAHN | N/A | N/A | 0.119±0.005 | 0.780±0.015 |
| ChemRL-GEM | 0.849±0.003 | 0.742±0.004 | N/A | N/A |
| GraphGPS | 0.849* | 0.719* | 0.087(0.133)* | 0.788±0.010 |
| Ours | 0.841±0.003 | 0.733±0.001 | 0.143±0.026 | **0.795±0.008** |
| Ours(dual readout,VN) | 0.844±0.006 | 0.744±0.010 | 0.160±0.014 | 0.781±0.010 |
| Ours(dual readout) | 0.851±0.008 | **0.752±0.003** | **0.182±0.019** | 0.756±0.007 |

**Long-range interactions of molecular clusters.** The SubFormer architecture is designed to capture molecular properties that depend on both local and non-local interactions among atoms in a molecule. A widely adopted strategy to capture long-range interactions is to add a virtual node connected to all original nodes in the graph [16, 20]. In the example in Figure 1B, five MP layers are needed to transmit information from the rightmost node to the leftmost node in the absence of the virtual node; the introduction of the virtual node acts as a shortcut, effectively reducing the distance between any pair of nodes to two hops at most. SubFormer obviates this strategy. First, coarsening the graph into a junction tree reduces the number of hops between nodes, especially for molecules that contain multiple rings. Then the self-attention mechanism allows tokens of atom clusters to interact even when they are not directly connected in the coarse-grained tree. As indicated in Tables 1 and 3, adding a virtual node (VN) to the graph does not benefit the SubFormer approach.

**SubFormer attention weights correspond to chemically meaningful fragments.** To gain insight into how SubFormer functions, we plot the coarse-grained attention weights for representative molecules of the ZINC dataset in Figs. 2 and 7; the colored coarse-grained tree and molecular graphs

show the attention weights learned by the classification token from the last layer. There is a clear correspondence between the attention weights and the chemical structures. In particular, we see that aromatic rings generally have high attention weights. The sparsity of high attention weights for the classification token indicates the model's ability to focus on specific molecular fragments.

To ensure these results were not specific to the ZINC benchmark, we also trained SubFormer to predict the energy gap of frontier orbitals generated by density functional theory using the organic donor-acceptor molecules dataset of the computational materials repository [33]. In this dataset, the donors are typically conjugated and aromatic systems, while the acceptors are typically highly electronegative functional groups, such as ones containing fluorine. SubFormer achieves chemical accuracy of 0.07 eV on the hold-out test set, which is less than the typical error of the density functional theory calculations [36]. In this case, the SubFormer attention weights often correspond to molecular fragments participating in charge-transfer, as illustrated in Fig. 2. An additional example of SubFormer capturing long-range charge-transfer in a large molecular system is depicted in Fig. 8. Such long-range interactions are very challenging for conventional MPNNs to capture.

In Figs. 2, 7 and 8, we also present attention maps averaged across the attention heads. While the self-attention mechanism centers on particular nodes, the averaged attention maps become progressively flatter (note the varying scales of the heatmaps). This observation prompted us to further probe SubFormer's over-smoothing behavior.

**Over-smoothing and over-squashing in SubFormer: Analysis of the ZINC dataset.** Graph neural networks, while powerful, can exhibit over-smoothing, especially when deeper architectures are employed. Here, we explore the behavior of SubFormer in this regard. We specifically used SubFormer(Slim) with the dual output mechanism trained on the ZINC dataset (Table 1). With fixed feature dimension, we gradually increase the depth of SubFormer(slim) by stacking additional encoder layers. Usually, a performance increase would be expected as more parameters are introduced. However, as illustrated by Fig. 3, the accuracy clearly diminishes beyond three transformer encoder layers, which we attribute to over-smoothing based on the analysis above. To verify that this is indeed the case, we use the Dirichlet energy:

$$\mathcal{E}(X^{(l)}) = \frac{1}{n} \sum_{i \in V(G)} \sum_{j \in N(i)} \|X_i^{(l)} - X_j^{(l)}\|_2^2, \tag{8}$$

where $X^{(l)}$ represents the input to the $l$-th layer. Because $\mathcal{E}(X^{(l)}) \to 0$ if and only if the node features are uniform [29], this metric can be used to quantify over-smoothing.

We see in Fig. 3 that the Dirichlet energy decreases for SubFormer as the layer index increases, although the change is much more gradual than for GAT. By contrast, GraphGPS maintains a consistently high Dirichlet energy, indicating that it does not suffer from over-smoothing. GraphGPS achieves this behavior by interleaving MP and self-attention. In effect, GraphGPS uses two distinct adjacency matrices: its MP component utilizes the graph adjacency matrix to aggregate information from nearest neighbors, while its self-attention component exchanges information in a fully-connected fashion. This interplay suppresses over-smoothing. However, interleaving MP and self-attention is difficult to justify mathematically and complicates interpretation [1, 43].

As noted previously, over-squashing is another potential issue which arises in MPNNs applied to graphs with long-range interactions. To assess the significance of this issue for chemical applications, we plot the distribution of shortest path lengths to a reference node (the starting node selected by the force-directed graph drawing algorithm as implemented in the NetworkX package [18]) for several datasets in Fig. 6. We see that paths with 15 or more hops frequently arise, necessitating a comparable number of MP layers, which should lead to over-squashing.

We quantify over-squashing by computing the Jacobian $\partial x_i^{(l)}/\partial x_j$, where $x_j = x_j^{(0)}$ is the initial feature of a distant node $j$ relative to $i$. Tending to zero means the input of node $j$ loses its influence on node $i$, and we can attribute values close to zero to over-squashing when under-reaching is not an issue due to the number of MP layers [46, 8].

To illustrate this phenomenon, we picked a subset of molecules from the ZINC dataset and computed the Jacobian for GAT and SubFormer. For the latter, we did not read the classification token to be consistent with GAT. We furthermore did not use the dual readout to prevent the model from accessing the original graph information directly. Instead, we mapped the coarse-grained feature matrices back to the original graph. Then, we computed the Jacobian of the feature vector of the reference nodes

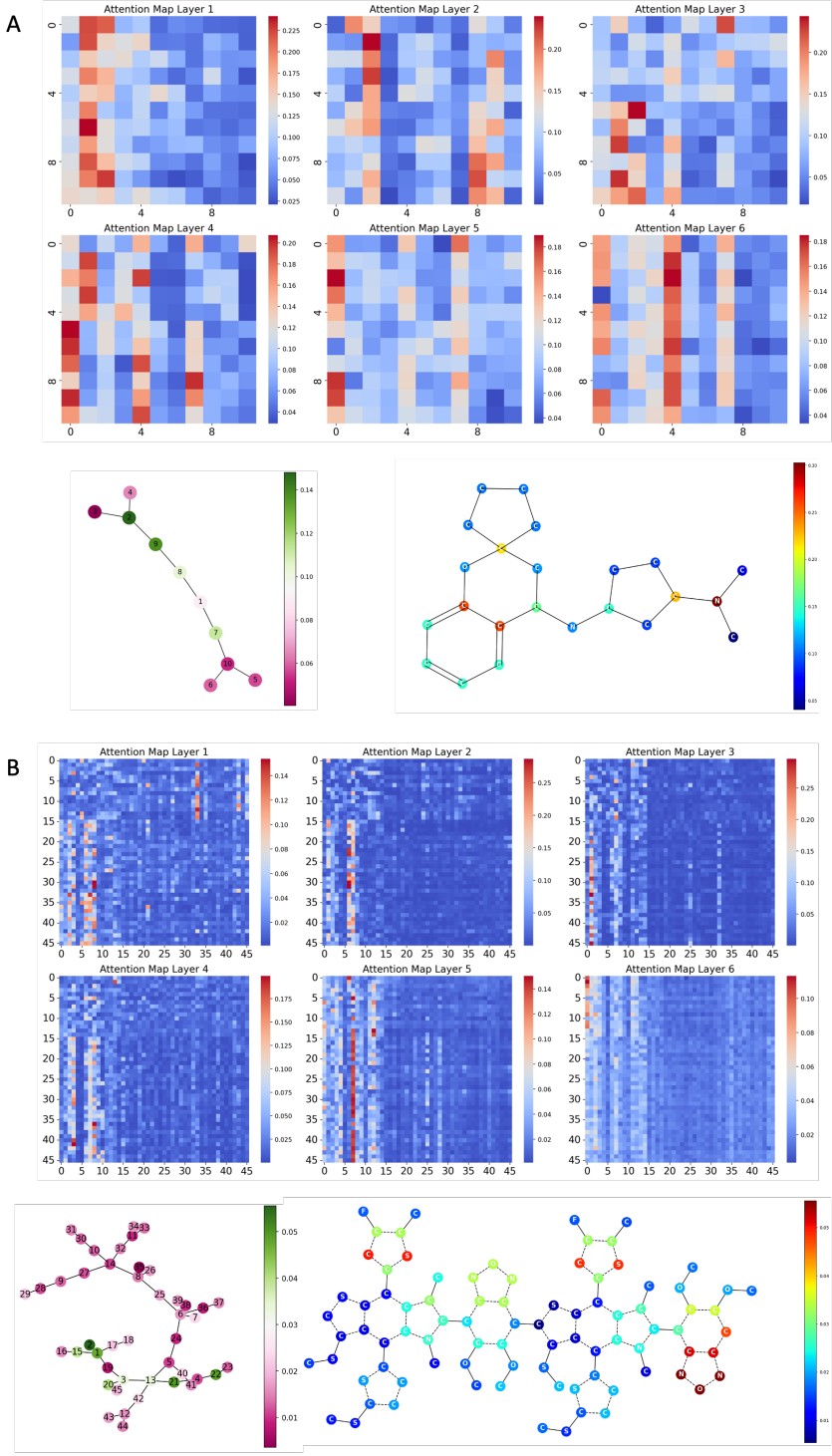

Figure 2: Attention maps (heat maps) for representative molecules from (A) the ZINC dataset (B) the organic donor-acceptor dataset (for clarity, aromatic systems are labeled with dashed lines). Node 0 is the classification token. The molecular graphs and trees with corresponding scale bars show the attention weights learned by the classification token.

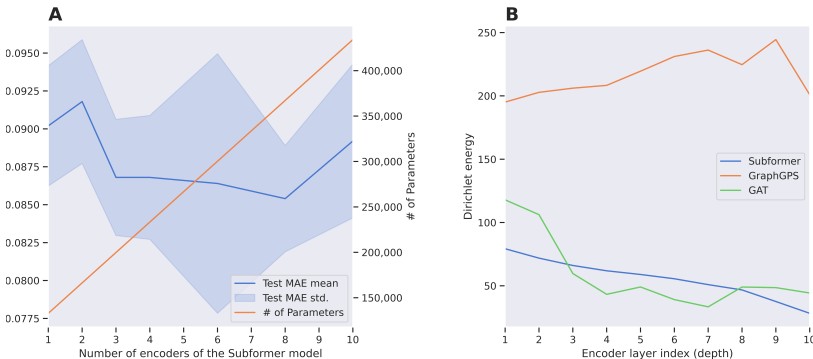

Figure 3: Increasing the number of layers leads to over-smoothing. (A) Dependence of performance on the number of transformer encoder layers for SubFormer(slim) applied to the ZINC dataset. (B) Comparison of the Dirichlet energy at different encoder layers among SubFormer, GraphGPS, and GAT applied to the ZINC dataset.

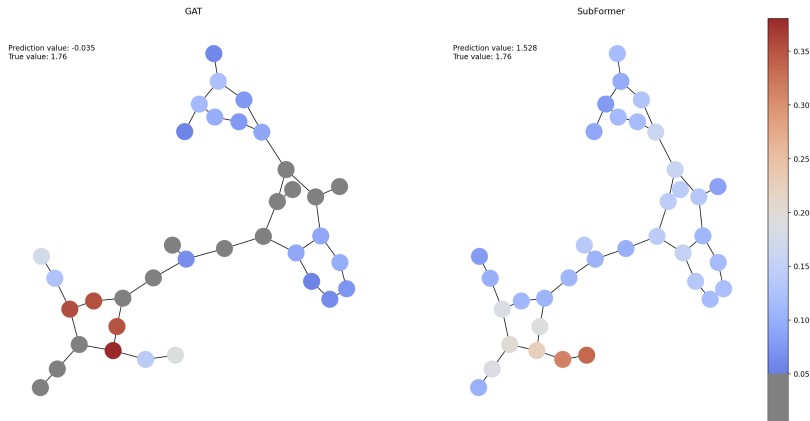

Figure 4: Norm of the Jacobian with respect to the input features. Nodes with gradient norm less than 0.05 are highlighted dark gray.

used above with respect to all input features and took the norm. We show representative results in the Fig. 4. We color the nodes with a gradient norm less than 0.05 dark gray. As described in section 2, GAT exhibits over-squashing in larger graphs with branches, while SubFormer does not.

## 4   Conclusions

Efficient graph learning paradigms have been a focal point for both the machine learning and chemistry communities. In this study, we introduced SubFormer, which combines hierarchical clustering of nodes by tree decomposition and MP with a transformer architecture for learning both local and global information in graphs. Our approach is motivated by the longstanding practice of interpreting chemical structures in terms of functional fragments. Here we showed that decomposition of graphs into fragments is not just useful for interpretation but also addresses graph representation learning challenges and reduces computational cost. SubFormer performed consistently well on standard benchmarks for predicting molecular properties from chemical structures. Visualization of the attention maps suggests that the model learns chemically pertinent features. We showed that SubFormer does not suffer from over-squashing because the attention weights obviate extensive MP to capture long-range interactions. Over-smoothing, while better controlled than in some graph transformers, remains an issue, demanding further study. Nevertheless, our results demonstrate that standard transformers, when equipped with local information aggregated via message-passing, excel as hierarchical graph learners.

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

# A Weisfeiler-Lehman Test and Expressive Power of SubFormer

A central goal of graph learning models is to distinguish non-isomorphic graphs. Two graphs are isomorphic if their nodes and edges are in one-to-one correspondence. Despite this simple description, in the language of complexity theory, the so-called graph isomorphism problem is neither known to be NP-complete nor known to be polynomial-time solvable [24]. Weisfeiler-Lehman algorithm provides a weaker but possibly efficient approach to separate part of graphs. Let $\chi : V(G) \to \mathbb{N}$ be a function that colors each node of the graph $G$ with a natural number. Then the 1-WL algorithm refines color as follows:

$$
\begin{aligned}
\chi^{(0)}(v) &= \chi(v), \\
\chi^{(l+1)}(v) &= (\chi^{(l-1)}(v), \{\!\{(\chi^{(i-1)}(w)); w \in N_G(v)\}\!\})
\end{aligned}
\tag{9}
$$

where $\{\!\{(\chi^{(i-1)}(w)); w \in N_G(v)\}\!\}$ denotes a multiset. Values of $\chi^{(l+1)}$ are then sorted lexicographically. Fig. 5 shows an illustration of the 1-WL algorithm. Definition of higher-order WL tests can be found in [4, 21, 3]. Even though almost all graphs can be separated by color refinement [5, 32], there exist counterexamples even among small molecular graphs like those in Fig.5. Graph representation learning algorithms that scale subquadratically with node number, like MPNN [16], are at most as expressive as the 1-WL test [54, 3, 27].

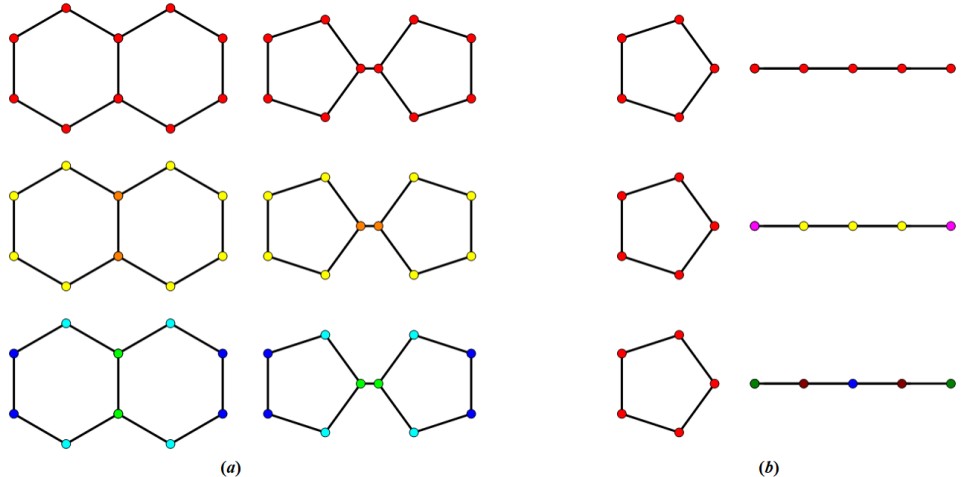

$(a)$                 $(b)$

Figure 5: Demonstration of the 1-WL algorithm (color refinement) using Eq. (9). (a) Comparison of two interconnected six-member rings with two interconnected five-member rings sharing a single node. (b) Comparison of a five-member ring with a line of five nodes. While the 1-WL algorithm can differentiate the structures in (b), it fails to distinguish those in (a), as indicated by the node colors in the final row.

**Theorem A.1.** *SubFormer is more expressive than 1-WL in testing non-isomorphic graphs.*

*Proof.* SubFormer is a hybrid of a MPNN and a transformer. A MPNN has an expressivity that is equivalent to the 1-WL algorithm [54, 3], while a standard transformer with appropriate positional encoding and augmentation on tokens is at least as expressive as the 1-WL test [27]. The 1-WL test thus represents a lower bound on expressivity.

Consider non-isomorphic graphs $G_1$ and $G_2$ with trees $T_1$ and $T_2$ obtained from a graph decomposition scheme. Any two non-isomorphic trees are distinguishable under color refinement, or equivalently the 1-WL test [21, 41]. SubFormer can thus distinguish $G_1$ and $G_2$ if $T_1$ and $T_2$ are non-isomorphic. Using the procedure to craft junction trees, Fig. 5 $(a)$ illustrates one case which can be separated by SubFormer but not by the 1-WL algorithm.     $\square$

Existing neural network architectures, like order-$k$ linear GNNs [37] or order-$k$ Folklore GNNs [38], achieve higher expressive power using higher-order tensors to encode graph features. We instead choose to compress graphs into local clusters and utilize a transformer to distinguish junction trees,

which avoids introducing more complicated tensor inputs. As a result, the separation power of our framework is limited by the method used to compress local structures of the original graph and we leave it to the future to further refine the decomposition strategy.

## B  Datasets

Table 4 provides an overview of the benchmark datasets used in this study, with a primary focus on the ZINC [22] and MoleculeNet [52] datasets. For consistency, we employed a data division ratio of 8:1:1 for training, validation, and testing, as recommended by [52]. Notable exceptions are the ZINC, MOLHIV, and Peptides-struct datasets, which have predefined split specification. These predefined ratios were sourced from their implementations in the PyTorch Geometric package [14] and the Open Graph Benchmark package [20]. Additionally, we depend on these packages for loss computation and to access standard node and edge features.

Table 4: Benchmark datasets used in the study. # Excluded is the number of samples that was excluded due to failure of the default tree decomposition scheme.

| Dataset | # Samples | # Tasks | Task type | Metric | Split | # Excluded |
|---------|-----------|---------|-----------|--------|-------|------------|
| ZINC | 12,000 | 1 | Regression | MAE | Random | 0 |
| Peptides-Struct | 15,535 | 11 | Regression | MAE | Stratified | 0 |
| TOX21 | 7,831 | 12 | Classification | ROC-AUC | Random | 19 |
| TOXCAST | 8,575 | 617 | Classification | ROC-AUC | Random | 11 |
| MUV | 93,087 | 17 | Classification | AP | Random | 0 |
| MOLHIV | 41,127 | 1 | Classification | ROC-AUC | Scaffold | 0 |

## C  Hyperparameters

The hyperparameters applied to all benchmark datasets in this study are detailed in Table 5. We did not perform a systematic search of hyperparameters due to limited computational resources. Notably, while the CosineAnnealingLR scheduler is frequently associated with the ZINC dataset, it was not employed in this research. Similarly, the GeLU activation function [19], often utilized for various transformers, was not incorporated. The local MP block, in its design, has the flexibility to integrate diverse MPNNs; however, this work specifically considered GINE [55] and the anti-symmetric variant [17] of GATv2 [6]. Tuning of hyperparameters could enhance performance.

Table 5: Hyperparameter settings used for the benchmarks. The dimension of the hidden features of the readout block depends on whether the dual readout is enabled. ROP stands for the ReduceLROn-Plateau scheduler, LPE stands for the graph laplacian matrix eigenvectors, and SPDE stands for the shortest path distance matrix eigenvectors.

| Model Comp. | Parameters | ZINC | TOX21 | TOXCAST | MUV | MOLHIV | Peptides-struct |
|---|---|---|---|---|---|---|---|
| Optimization | Epoch | 500 | 50 | 100 | 20 | 30 | 100 |
| | Learning rate | 0.001 | 0.0001 | 0.0001 | 0.0001 | 0.0001 | 0.0005 |
| | Optimizer | Adam | AdamW | AdamW | AdamW | Adam | AdamW |
| | Schduler | ROP | None | None | None | ROP | ROP |
| | Batch size | 64 | 32 | 64 | 32 | 32 | 64 |
| Local MP | # Layers | 2 | 3 | 8 | 5 | 3 | 2 |
| | # Hidden Features | 64 | 256 | 128 | 128 | 64 | 64 |
| | MP Type | GINE | GINE | aGATv2 | aGATv2 | GINE | GINE |
| | Aggregation | Sum | Sum | Sum | Sum | Mean | sum |
| | Activation | ReLU | ReLU | ReLU | ReLU | ReLU | ReLU |
| | Tree Activation | LeakyReLU | LeakyReLU | LeakyReLU | LeakyReLU | LeakyReLU | None |
| | Dropout | 0 | 0.2 | 0 | 0 | 0.05 | 0.05 |
| | Edge Dropout | 0 | 0.2 | 0.5 | 0.5 | 0 | 0.05 |
| Pos. Enc. | Encoder PE Emb. Dim. | 64 | 256 | 128 | 128 | 64 | 64 |
| | PE Dim. | 10 | 10 | 10 | 10 | 10 | 32 |
| | PE Type | DEG,SPDE | DEG,SPDE | DEG,LPE | DEG,LPE | DEG,SPDE | DEG,LPE |
| | PE Merge | Concat | Concat | Sum | Sum | Concat | Concat |
| | MP PE | None | LPE | None | None | LPE | None |
| Transformer | # Hidden Features | 128 | 512 | 128 | 128 | 128 | 128 |
| | # FFN Hidden Features | 128 | 1024 | 512 | 512 | 256 | 128 |
| | # Layers | 3 | 4 | 4 | 4 | 5 | 3 |
| | # Heads | 8 | 8 | 16 | 16 | 5 | 8 |
| | Activation | ReLU | ReLU | ReLU | ReLU | ReLU | ReLU |
| | Dropout | 0.1 | 0.5 | 0.2 | 0.2 | 0.5 | 0.05 |
| | Padding Dim. | 40 | 120 | 120 | 50 | 260 | 530 |
| Readout | # Hidden Features | 192/128 | 768/512 | 256/128 | 256/128 | 256/128 | 128 |
| | Activation | ReLU | ReLU | ReLU | ReLU | ReLU | ReLU |

# D Timing

We report the computional time per epoch and the total number of epochs required for each dataset in Table 6. All training was done on a single Nvidia RTX 3080 card with single-precision float format. Automatic mixed precision (AMP) could accelerate the training over 50%. The numbers reported are without AMP at torch.float32 format except for the Peptides-struct benchmark. Theoretically, SubFormer could be further accelerated by incorporating other common techniques such as PerFormer [7], Linear Transformer [25], etc.

Table 6: Training time per iteration and total number of iterations required to complete training.

| Dataset | Time per epoch(sec) | Num. of epochs |
|---|---|---|
| ZINC | 3 | 500 |
| TOX21 | 4 | 40 |
| TOXCAST | 5 | 100 |
| MUV | 36 | 20 |
| MOLHIV | 45 | 30 |
| Peptides-struct | 9 | 100 |

# E Additional analysis

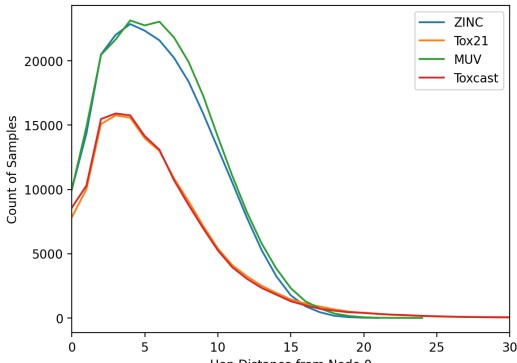

Figure 6: Distribution of hopping distances from a reference node (the starting node selected by the force-directed graph drawing algorithm as implemented in the NetworkX package [18]) to all other nodes in the indicated datasets.

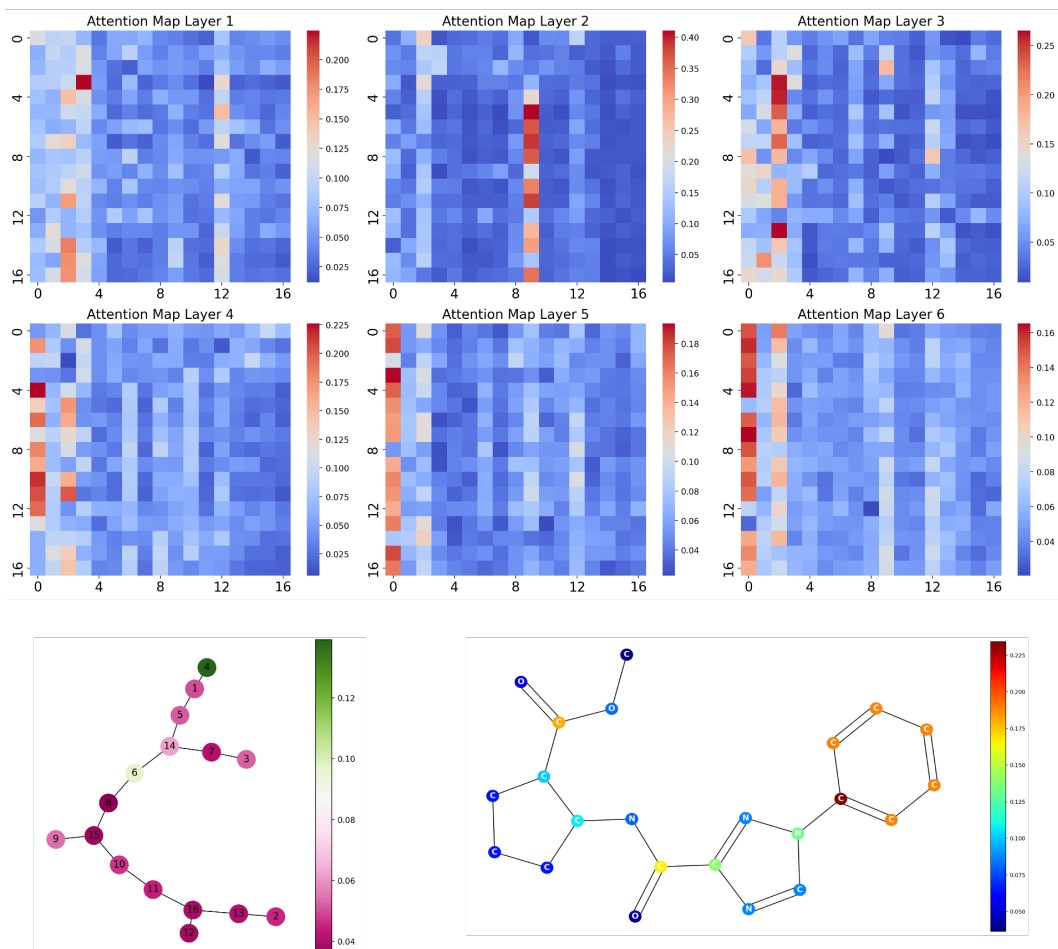

Figure 7: Same as Fig. 2 for an additional molecule from the ZINC dataset

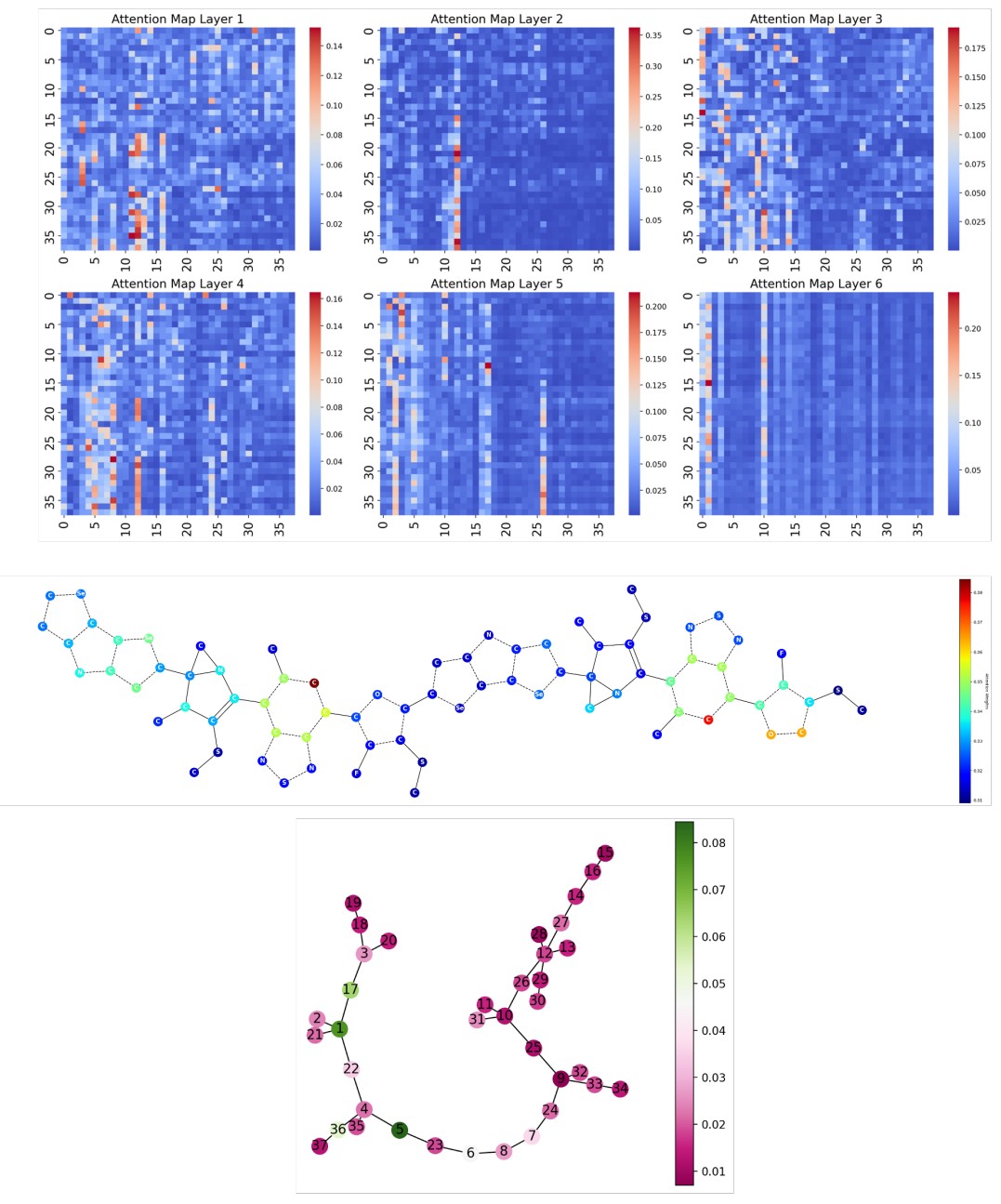

Figure 8: Same as Fig. 2 for an additional long-chain molecule from the organic donor-acceptor dataset

