# OpenReview forum: "Transformers are efficient hierarchical chemical graph learners"
_NeurIPS.cc/2023/Workshop/AI4Science — NeurIPS2023-AI4Science Poster_

### Official Review · Reviewer_Fk8t · 2023-10-23
**Interesting and simple idea with potential for further empirical exploration**

**Rating:** 6
**Confidence:** 4

**Review:**

## Summary
This paper introduces a novel molecular graph representation learning architecture called SubFormer that combines the advantages of message-passing neural networks (MPNNs) with Graphs Transformers to decrease computational complexity and mitigate issues such as over-smoothing and over-squashing. In the first stage of the SubFormer, local graph-level information is aggregated using a shallow MPNN and compressed into a coarse-grained junction tree. In the second stage, a Graph Transformer is applied to the coarse-grained representation to enable the learning of long-range interactions. This also reduces the computational cost of applying the Graph Transformer directly to the input graph.

The authors demonstrate that the SubFormer’s performance is competitive with GraphGPS and outperforms other Graph Transformer and MPNN architectures on molecule property prediction benchmarks. They claim that this performance is achieved at a fraction of the computational cost of the other architectures. The authors also show that the attention weights on the junction tree can be interpreted as chemically meaningful fragments.

## Strengths
1. The idea is simple and intuitive.
2. The paper tackles two important problems simultaneously: computational efficiency and learning long-range interactions on graphs.
3. The analysis of the attention weights confirms that the model learns representations that correspond to chemical intuition.

## Weaknesses

1. The authors claim that the SubFormer is able to achieve comparable performance to SOTA Graph Transformer architectures on molecular graph learning tasks at a fraction of the computational cost, which is the main benefit over GraphGPS. This makes sense intuitively since the transformer is not applied to the entire input graph and the MPNNs are shallow. However, the decrease in cost is not quantified in the paper. The authors are encouraged to report the model complexity/training time for other architectures in comparison to SubFormer to empirically verify the claim.

2. Over-smoothing remains an issue for SubFormer while it is not prevalent in GraphGPS. Do the authors foresee possible ways to address this issue in the future without sacrificing interpretability as in GraphGPS?

3. The benefit of Graph Transformers as opposed to other architectures for learning on the junction tree is unclear besides the intuition that it provides a shortcut for learning long-range interactions by operating on a fully-connected graph. An interesting experiment would be to keep stage 1 of the SubFormer fixed while using a virtual node instead of the transformer (as it is currently only tested as an additional layer) or trying different MPNN architectures such as GATs.

## Minor comments
* In line 46, it is more accurate to say that the “expressive power” of graph representation learning models is assessed in terms of their ability to separate non-isomorphic graphs rather than “performance”, which is generally task-dependent.
* In the general MPNN update recursion formula, the learnable functions $\Psi$ and $\Phi$ also depend on the layer $l$ as $\Psi^{(l)}$ and $\Phi^{(l)}$.
* The font size in Figure 4 is too small to read easily.

## Final assessment
The proposed approach to combine GNNs and Transformers is simple and achieves good empirical results on various molecular graph learning benchmarks. I liked the comment on the contrast between GNNs and Graph Transformers in their use of structural inductive bias, which warrants investigation. The empirical analysis could be solidified by quantifying the decrease in the computational cost of SubFormer in comparison to SOTA molecular graph learning architectures, as well as by isolating the contributions of the Transformer architecture in stage 2.

---

### Official Review · Reviewer_PJuH · 2023-10-25
**Innovative idea, but the paper needs more organization**

**Rating:** 6
**Confidence:** 3

**Review:**

Summary:

The paper tackles the problem of molecular structure analysis and properties prediction for chemical research fields. Motivated by the intricate data structure of molecules, the limited learning capacity of graph networks, and the computational complexity of transformers, the authors propose to make the best out of both worlds and propose a hybrid approach that takes advantage of the graph learning capabilities of GNN and Transformers. Additionally, the proposed approach incorporates a hierarchical clustering method that reduces the number of graph nodes and elaborates a coarse-grained graph representation.

The idea of the paper is innovative and well-established. Combining GNN, clustering, and Transformers is a smart strategy to mitigate the inefficiency of Transformers in learning graph properties and limitations of GNNs in capturing long-range dependencies between nodes. The proposed approach has also been validated on different datasets, and the results were thoroughly discussed and analyzed.

One critical problem for me is the paper organization. For example, Section 3 should be named Methodology and comprise the paragraphs that describe the approach. Section 2 can be merged with the methodology part to give a harmonized section that details the idea. Another problem is the lack of complexity assessment. The motivation for not relying on Transformers only for these types of tasks is the computational complexity as the number of tokens increases drastically. However, the evaluation section doesn't provide any further details on the complexity of the models (the authors' models or SOTA models). The computational complexity can be either evaluated using proxy metrics (e.g., number of FLOPS and parameters) or directly by measuring execution time and memory utilization. This part is missing in this paper and needs to be included, especially as the authors employ two computational intensive models in their approach – GNN and Transformers.

The paper's Figures should be enhanced. For example, Figure 1 is not organized with different parts, and notations are barely visible. Titles, labels, and notations in all figures are very small and not readable at all.

Overall, the paper tackles a relevant and critical scientific problem in molecular analysis. The proposed idea seems promising. However, the only problem for me is the paper organization which needs to be improved in future revisions.


Strengths:
- The paper tackles an important problem in molecular analysis.
- The idea of combining GNN, Transformers, and clustering is well-motivated and promising.
- The approach has shown impressive results on different datasets.


Weaknesses:
- The evaluation section lacks a discussion on the computational complexity of the obtained models and state-of-the-art models. A comparison should be provided to draw a concrete conclusion on the effectiveness and efficiency of the proposed approach.
- The paper needs more organization. The jump from Section 2: Background to Section 3: Results is unorthodox.
- In equation (4), the difference between the trainable weights of the junction tree and the Transformer should be explicitly explained in the text. Also, notations should be unified for the MPNN line and the last two lines.
- The use of junction trees is not well justified. What are the other alternatives? Why is this a better choice for this specific case?
- Figures need to be enhanced and nicely presented.

---

### Meta-Review · Area_Chair_i2YR · 2023-10-27

**Recommendation:** Accept (Poster)
**Confidence:** 4

**Metareview:**

The paper delves into an intriguing methodology for molecular structure analysis, combining the strengths of message-passing neural networks (MPNNs) and Graph Transformers to optimize molecular graph representation learning. This hybrid approach, termed SubFormer, seeks to counteract the challenges posed by the complexity of molecular data, the limitations of graph networks, and the computational intensity of transformers.

Strength:
            At the heart of the paper is the novel two-stage process. In the initial stage, the paper details how local graph information is aggregated using MPNNs, condensed into a more manageable junction tree. Subsequently, a Graph Transformer is applied to this compressed representation to facilitate the learning of longer-range interactions, aiming for improved computational efficiency.

While the paper boasts competitive results with SubFormer, outpacing several state-of-the-art architectures on molecule property prediction benchmarks, there are areas where it could be further fortified:

* Computational Efficiency: A primary motivation for the proposed methodology is the computational challenges of transformers, especially with escalating token numbers. However, the paper stops short of quantifying this improved efficiency, leaving readers without a concrete comparison. It would strengthen the paper to provide metrics—be it the number of FLOPS, parameters, execution times, or memory utilization—to quantify and validate these efficiency claims.

* Organization & Presentation: The paper's structure appears somewhat disjointed, with an unexpected progression of sections. Furthermore, the visual components, such as figures, would benefit from a revamp for clarity and legibility.

* Depth in Methodology: While the combination of GNNs, Transformers, and clustering is commendable, certain aspects, like the justification for using junction trees or the distinct trainable weights, require further elaboration. Exploring alternatives or providing a more detailed rationale would offer readers deeper insights.

* Smoothing Concerns: Over-smoothing, a prevalent issue in the domain, isn't thoroughly addressed for SubFormer. Potential future strategies to tackle this, without compromising other aspects, would be a valuable addition.

* Empirical Analysis Augmentation: The empirical evidence provided is compelling but could be solidified by isolating specific contributions, especially that of the Transformer in the second stage. Furthermore, nuances such as the expressive power versus performance delineation would provide greater clarity.